# IDNet: An inception-like deformable non-local network for projection compensation over non-flat textured surfaces

**Yuqiang Zhang, Huamin Yang***, **Cheng Han, Chao Zhang, Chao Xu, Shiyu Lu**

School of Computer Science and Technology, Changchun University of Science and Technology, Changchun, Jilin, China

* yanghuamin@cust.edu.cn

**Data availability statement:** The dataset includes images not collected or owned by the authors which are subject to copyright

## Abstract

Projector compensation on non-flat, textured surfaces represents a formidable challenge in computational imaging, with conventional convolution-based methods frequently encountering critical limitations, especially in image edge regions characterized by complex geometric transformations. To systematically address these persistent challenges, we introduce *IDNet*, an innovative framework distinguished by its multi-scale receptive feature extraction modules. Central to our approach are multi-scale deformable convolution modules that dynamically adapt to geometric distortions through intelligently flexible sampling positions and precise offset mechanisms, which significantly enhance processing capabilities in intricate distortion regions. By strategically integrating non-local attention mechanisms, *IDNet* comprehensively captures global contextual information, thereby substantially improving both geometric and photometric compensation accuracy. Our experimental validation demonstrates that the proposed method achieves comparable compensation performance to existing approaches, particularly in the most challenging and geometrically complex edge regions of projected images.

## 1 Introduction

Projector compensation has emerged as a fundamental technique in projection mapping and display systems, addressing the inherent challenges of achieving accurate and high-quality projected imagery on non-ideal surfaces. This technology has become increasingly important as projection systems expand beyond traditional flat screens to encompass complex geometric surfaces, textured materials, and dynamic environments.

Traditional approaches to projector compensation can be broadly categorized into two main methodologies: geometry-based compensation and photogeometric compensation. Geometry-based compensation focuses on correcting spatial distortions caused by surface irregularities and projection angles, typically employing structured light patterns and camera feedback systems to construct accurate surface models. This approach has proven highly

restrictions. The authors have provided the URLs to the third party image sources in the Supporting Information, so others may request permission from the copyright holders to access and use the images. The authors confirm they did not have any special access privileges that others would not have.

**Funding:** This research was funded by Jilin Province Science and Technology Development Plan Project, grant number: 20230101179JC, National Natural Science Foundation of China Youth Fund Project, grant number: 61702051, Jilin Province Science and Technology Development Plan Project, grant number: 20200403188SF. The funders had no role in study design, data collection and analysis, decision to publish, or preparation of the manuscript.

**Competing interests:** The authors have declared that no competing interests exist.

effective for static environments but faces limitations in dynamic scenarios. Photogeometric compensation, on the other hand, addresses the color and intensity variations introduced by surface properties and environmental lighting. This method typically involves creating a detailed model of the projection surface's reflectance properties and environmental lighting conditions to adjust the projected image accordingly. While geometric and photogeometric compensation can significantly improve image quality on non-white or textured surfaces, it often requires complex calibration procedures and may struggle with highly specular or dark surfaces where the projector's dynamic range becomes a limiting factor.

Recent advances in projector compensation have introduced hybrid approaches that combine both geometric and photometric compensation, often leveraging machine learning techniques to improve accuracy and visual effectiveness. These end-to-end methods [1–4] aim to simultaneously address geometric and photometric distortions, providing a more comprehensive solution for projection on non-flat textured surfaces. However, existing methods often face challenges in capturing complex geometric transformations and global contextual information, particularly in edge regions where image distortions are most pronounced. To overcome the challenges observed in the previous methods, our proposed compensation network, integrating Resblock [5], Inception-like convolution, and non-local attention, addresses these limitations. This design captures comprehensive image context by extracting global features and effectively integrating them with local features to learn rich contextual information. Overall, our contributions in this paper can be summarized as follows:

- We present an innovative compensation framework that jointly addresses geometric and photometric distortions in projection systems through a unified network architecture. At its core, the framework leverages three specialized Inception-inspired modules, strategically designed to enhance multi-scale feature learning.
- We conduct comprehensive comparative evaluations using both standard benchmark datasets and our newly curated datasets, providing detailed insights into the method's capabilities for handling the complex challenges that arise in real-world projection environments.
- To overcome the limitations of existing datasets, we systematically curated a comprehensive collection of 5,000 diverse projection samples, which substantially improved our network's generalization capabilities and computational robustness across varied projection scenarios.

## 2 Related work

Projection technology has undergone a remarkable transformation since its inception, evolving from simple display systems into sophisticated tools for spatial computing and augmented reality. This evolution began with pioneering research by Raskar et al. [6], who established the foundational principles of spatial augmented reality and opened new possibilities for human-computer interaction. As the field matured, researchers expanded these concepts into diverse applications, developing advanced techniques for projection stereo [7–9], creating responsive interactive displays [10,11], and designing immersive environments that blend digital content with physical spaces [12]. The technical challenges inherent in projection systems have sparked parallel research streams: one significant direction focuses on addressing projection blur, which arises from both optical and digital sources, through innovative applications of convolutional neural networks [13–15]. Equally important has been the advancement of projection compensation techniques for non-flat textured surfaces, an area that has gained particular prominence in recent years as researchers work to overcome the complex geometric and photometric distortions inherent in real-world projection scenarios.

## 2.1 Traditional methods for projection compensation

Traditional methods of projecting images onto non-flat textured surfaces typically involve two separate steps. These approaches initially tackle the *geometric compensation*, followed by adjustments in image brightness and color to achieve *photometric compensation*.

In order to compensate for the distortion caused by non-planar surfaces, which is referred to as *geometric compensation*, a series of methods [16–19] are based on structural light information. They employed a point-to-point approach to establish a mapping relationship between camera images and projected images, aiming to rectify geometric distortions caused by the projection. However, their ability to correct complex curved surfaces is limited, as they can only address straightforward surface scenarios. Except the methods based on the structural light, the pre-deformation-based approaches [20–24] achieve compensation effect by extracting the geometric information of points on the projection surface and then carrying out a reverse mapping of the image. To counteract the effects of projection geometric distortion in dynamic videos, FM-OF [25] involves estimating the initial homography matrix. Subsequently, images are subject to geometric deformation through the computation of subtle variations between the two compensation frames. While these approaches do not necessitate structured light technology, their compensation outcomes prove inadequate when confronted with videos characterized by non-rigid motions.

Expanding on the established geometric compensation, it becomes viable to subsequently tackle the *photometric compensation*. Certain methods [26–28] compensate for projecting onto a textured surface by deriving a discrete response function through extracting corresponding pixels from both the projector and camera images. To achieve photometric compensation for textured Lambertian surfaces [29,30], a linear formula is utilized to compute the compensation image. Additionally, other approaches address photometric distortion [31–33] by employing a scaled luminance-ratio map with the estimated response function, enhancing the accuracy of projection compensation on surfaces with geometric distortion.

However, these traditional methods possess the disadvantages that they are not end-to-end designs and typically require sampling operations on the image, which can lead to edge distortions or blurry textures.

## 2.2 End-to-end projection compensation methods

To overcome the limitations of traditional projection compensation approaches, recent advancements have introduced end-to-end projection compensation methods. These methods have manifested in two primary forms: those focused solely on photometric correction [34], and those that combine both geometric and photometric correction [2–4]. These methods utilize convolutional neural networks (CNNs) to estimate the photometric compensation function implicitly. For example, CompenNet++ [2] is the first method to consider both geometric and photometric factors simultaneously in its end-to-end design. By jointly optimizing the two sub-convolutional networks, it achieves a more comprehensive and accurate projection compensation effect. Additionally, it proposes a more objective benchmark for evaluating full projector compensation. CompenNeSt++ [3] is an enhanced version of CompenNet++ with new proposed network architecture that it emerged the geometric refine net of two branches into one which shares the parameters of the input surface image and projected image to improve the efficiency. CompenHR [4] is the first high-resolution solution for the full compensation problem. It utilizes a pixel attention module to help capture fine-grained information. However, the use of shuffle/unshuffle operations in place of upsampling and downsampling improves efficiency at the cost of disrupting the global continuity of the original image information.

Moreover, it is widely recognized that compensation methods relying on CNNs face challenges due to the constrained receptive field imposed by convolutional kernel sizes, leading to an inadequate capture of global information [35]. Consequently, when dealing with surfaces exhibiting significant geometric distortion, compensation could result in edge artifacts, leading to errors in edge correction.

To overcome the restricted receptive field limitation inherent in CNNs, we have introduced an innovative network. This network combines multiscale feature extraction with deformable convolutions and non-local attention mechanisms. This novel approach is designed to comprehensively capture the global information present in the compensation image, thus achieving improved restoration of both texture and shape. Importantly, our method offers an end-to-end solution capable of addressing both geometric and photometric distortions.

## 3 Method

In this section, we first introduce the preliminary for the full projector compensation in Section 3.1. Then, we carefully design the module of *IDNet* in Section 3.2.

### 3.1 Problem formulation

The photometric and geometric compensation system comprises a stationary camera-projector pair utilized for both projection and capture. The surroundings where each projected image is captured maintain a condition of subdued and uniform environmental lighting. Based on the full compensation formulation proposed by Huang et al. [3], we denote the input image for the projector as $I : \mathbb{R}^{H_1 \times W_1 \times 3}$, the captured projected image as $\tilde{I}$, the combined attributes of surface reflectance, geometry, and pose as $S$, and the overall global lighting as $G$, the formulation for the captured projected image $\tilde{I}$ can be expressed as follows:

$$\tilde{I} = \mathcal{T}\big(\mathcal{F}(I; G, S)\big) \tag{1}$$

where the function $\mathcal{T} : \mathbb{R}^{H_1 \times W_1 \times 3} \mapsto \mathbb{R}^{H_2 \times W_2 \times 3}$ is responsible for geometrically warping the input image $I$ to the camera-captured image, while the function $\mathcal{F} : \mathbb{R}^{H_1 \times W_1 \times 3} \mapsto \mathbb{R}^{H_1 \times W_1 \times 3}$ photometrically transforms the input image $I$ into an uncompensated camera-captured image aligned with the projector canonical and frontal view.

The goal of photometric and geometric compensation is to obtain a compensated version $I^*$ of the projector input image $I$, which already corrects the photometric and geometric distortions caused by the projection environment. Then the ideal resulting image $I'$ perceived by a viewer is formulated as:

$$I' = \mathcal{T}\big(\mathcal{F}(I^*; G, S)\big) \tag{2}$$

where $I'$ is an affine transformed $I$, and consequently the $I^*$ in Eq 2 can be formulated as:

$$I^* = \mathcal{F}^\dagger\big(\mathcal{T}^{-1}(I'); \mathcal{T}^{-1}(\tilde{S})\big) \tag{3}$$

where $\mathcal{F}^\dagger$ is the pseudo-inverse of $\mathcal{F}$ and $\mathcal{T}^{-1}$ is the inverse geometric transformation of $\mathcal{T}$. Meanwhile, the $\mathcal{T}^{-1}(\tilde{S})$ in Eq 3 is the substitution of surface reflectance property $S$ and global lighting $G$. Similarly, from Eq 1, it can be deduced that:

$$I = \mathcal{F}^\dagger\big(\mathcal{T}^{-1}(\tilde{I}); \mathcal{T}^{-1}(\tilde{S})\big) \tag{4}$$

This equation indicates that a network can be trained over the image pairs and the surface image to approximate the Eq 4. Denoting the training image pairs $\chi = \{(\tilde{I}_i, I_i)\}_{i=1}^{N}$, the corresponding surface $\tilde{S}$ and the network as $\omega_\theta^\dagger$, where $\theta = \{\theta_\mathcal{F}, \theta_\mathcal{T}\}$ is the trainable photometric and geometric transform parameters, the network parameter $\theta$ can be approximated according to:

$$\theta = \arg\min_{\theta'} \sum_i \mathcal{L}(\hat{I}_i = \omega_{\theta'}^\dagger(\tilde{I}_i; \tilde{S}), I_i) \tag{5}$$

where the loss function $\mathcal{L}$ is calculated between the inferred image $\hat{I}_i$ and the projector input image $I_i$. According to Eq 3 and Eq 4, the trained network can be utilized to derive the compensation image denoted as $I^*$.

Our proposed method, denoted as *IDNet*, is an end-to-end projector compensation network encompassing two sub-networks: the geometric parameters refine network (GRN), symbolized as $\mathcal{T}^{-1}$, and the photometric compensation network (PCN), represented as $\mathcal{F}^{-1}$. The architecture is depicted in Fig 1. The GRN is designed to learn the deformed parameters of the sampled warping grid, while the subsequent PCN focuses on acquiring the features of both the geometrically warped surface image and the captured projected image. Nevertheless, in response to the restricted receptive field inherent in existing compensation networks [2,3], we undertake a redesign by incorporating larger kernel sizes into the networks and the detailed structure of *IDNet* is shown in Table 1.

The GRN comprises Downsample [36], DILM, NLM and TConv modules, as in Fig 1a. In the context of the GRN, the input affine grid and TPS grid [37,38] experience non-linear deformation. This specific form of deformation might not be suitably captured by conventional regular sampling techniques. Accordingly, we integrate multiscale deformable convolutions into the proposed ILM module, referred to as DILM, aimed at learning offset parameters for grid deformation. This augmentation markedly enhances the network's capacity to discern and accurately capture deformation-related details. The comprehensive design specifics

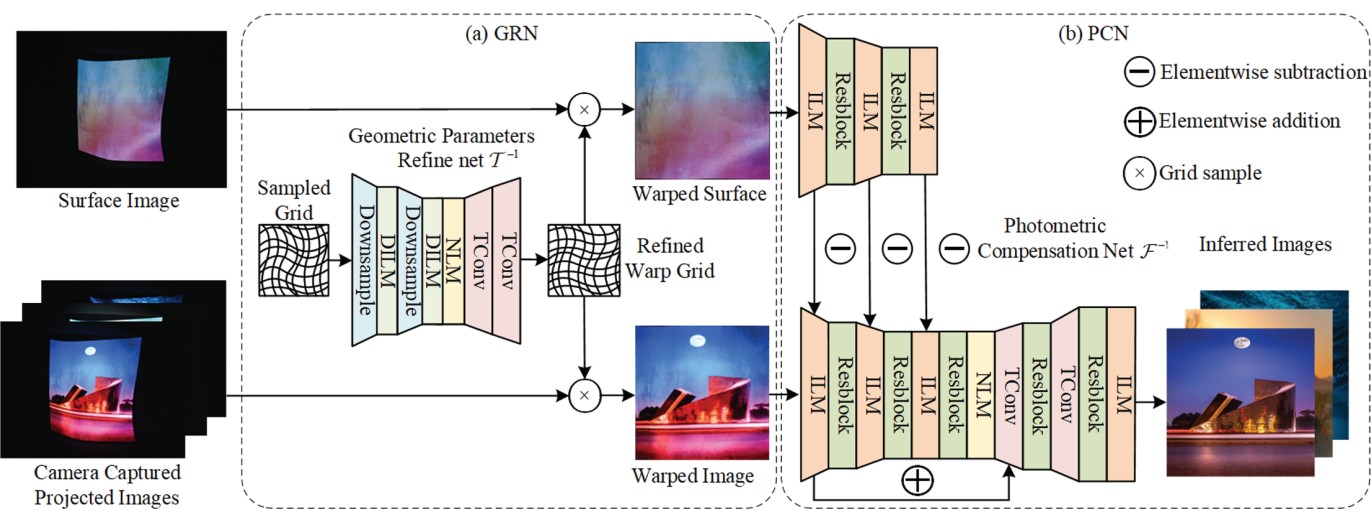

**Fig 1. The architecture of *IDNet* consists of two sub-networks: the Geometric Parameters Refinement net (GRN) optimizes the parameters of the warping grid, and the Photometric Compensation net (PCN) compensates for the photometric distortion caused by the surface and environment.** The DILM and NLM were used in the GRN to achieve a larger receptive field, aiding in the correction of geometric distortions. Meanwhile, the PCN employed ILM and NLM to acquire multiscale features and global information for photometric compensation.

**Table 1. Implementation details of the proposed network. (* Indicates the surface image and projected image encoder layers share the same design.)**

| IDNet | Layer | Output Size | Output Channel |
|---|---|---|---|
| GRN (Encoder) | Downsample | 128 | 32 |
| | DILM× 1 | 128 | 32 |
| | Downsample | 64 | 64 |
| | DILM×1 | 64 | 64 |
| GRN (Bottleneck) | NLM×1 | 64 | 64 |
| GRN (Decoder) | TConv×1 | 128 | 32 |
| | TConv×1 | 256 | 2 |
| PCN (Encoder) | * ILM×1 | 128 | 32 |
| | * Resblock×2 | 128 | 32 |
| | * ILM×1 | 64 | 64 |
| | * Resblock×2 | 64 | 64 |
| | * ILM×1 | 64 | 128 |
| | Resblock×6 | 64 | 128 |
| PCN (Bottleneck) | NLM×1 | 64 | 128 |
| PCN (Decoder) | TConv×1 | 128 | 64 |
| | Resblock×2 | 128 | 64 |
| | TConv×1 | 256 | 32 |
| | Resblock×2 | 256 | 32 |
| | ILM×1 | 256 | 3 |

of the DILM module are expounded in Section 3.2.2. Throughout the geometric correction procedure, leveraging global information aids in aligning essential geometric attributes within the feature image to ensure the persistence of structural and textural coherence. To achieve this, we introduce the NLM into the GRN's bottleneck. Ultimately, as the GRN concludes, transposed convolution operations are employed to upsample and generate the geometric correction grid.

Regarding the PCN, we adapt the encoder to accomplish multiscale feature extraction and projected image encoding, employing multiple Inception-like modules interconnected with residual modules. The PCN is composed of ILM, Resblock [5], NLM and transposed convolution operation(TConv) [36] modules, as shown in Fig 1b. Introducing the ILM module mitigates the constraint imposed by limited receptive fields by enabling the parallel utilization of multiscale convolutions for input feature extraction. Furthermore, prevailing full compensation methods face limitations in extracting global information. To address this without escalating network complexity, we incorporate the NLM into the PCN's bottleneck. The detailed design of ILM and NLM are described in Sec. 3.2.1 and Sec. 3.2.3. Additionally, the interaction between the surface and the projected image is realized through elementwise subtraction. Features extracted using an ILM with a shallow-layer skip connection are subsequently decoded through multiple transposed convolution operations and residual modules. This approach mitigates the issue of gradient vanishing during the training process and produces the fully compensated projected image output.

## 3.2 Module design of *IDNet*

In this section, we present three novel modules for the *IDNet* network: the ILM, the DILM, and the NLM. These modules are specifically designed to enhance the network's ability to capture multiscale features, adapt to geometric distortions, and incorporate global contextual information, respectively. Additionally, we conclude this section by introducing the loss functions used during the training process.

**3.2.1 ILM: inception-like module** The material and texture of the projection surface can significantly affect the lighting and color representation of the projected image, resulting in uneven brightness and contrast, as well as issues such as overexposure and excessively deep shadows in certain regions. Furthermore, the surface texture may interfere with the content of the projected image, increasing its visual complexity and making it more difficult to interpret. To effectively address these challenges, we have designed an Inception-like module (ILM) based on multi-scale convolution. Inspired by Inception and Residual networks [39,40], the ILM is proposed to enhance the receptive field of PCN. As shown in Fig 2(a), this module employs parallel 3×3, 5×5, and 7×7 convolutional operations, enabling it to extract diverse features across different receptive field sizes. The module uses smaller convolution kernels to capture fine details of the image, while larger kernels are applied to extract broader photometric variations, thus facilitating adaptive photometric compensation for the projected image.

In the task of photometric compensation, input images exhibit brightness variations across different scales, including overall scene brightness changes and local detail fluctuations. By incorporating multiscale feature extraction, the network's adaptability to varying brightness conditions is enhanced. Consequently, we integrate this module into the photometric compensation network, positioning it as a core component with the residual blocks of the encoder. Additionally, the ILMs is utilized to adjust the resolution and channels of the input features, enabling subsequent modules to capture both fine-grained and coarse-level information effectively.

**3.2.2 DILM: deformable Inception-like module** In existing full projection compensation techniques [2–4], the geometric transformations are performed using convolution operations with fixed sampling on regular convolution kernels. A convolution kernel $\mathcal{R}$ is used to sample the input feature map x, and the sampled values are then weighted and summed. For each location $p_0$ on the output feature map y,

$$y(p_0) = \sum_{p_n \in \mathcal{R}} w(p_n) \cdot x(p_0 + p_n) \qquad (6)$$

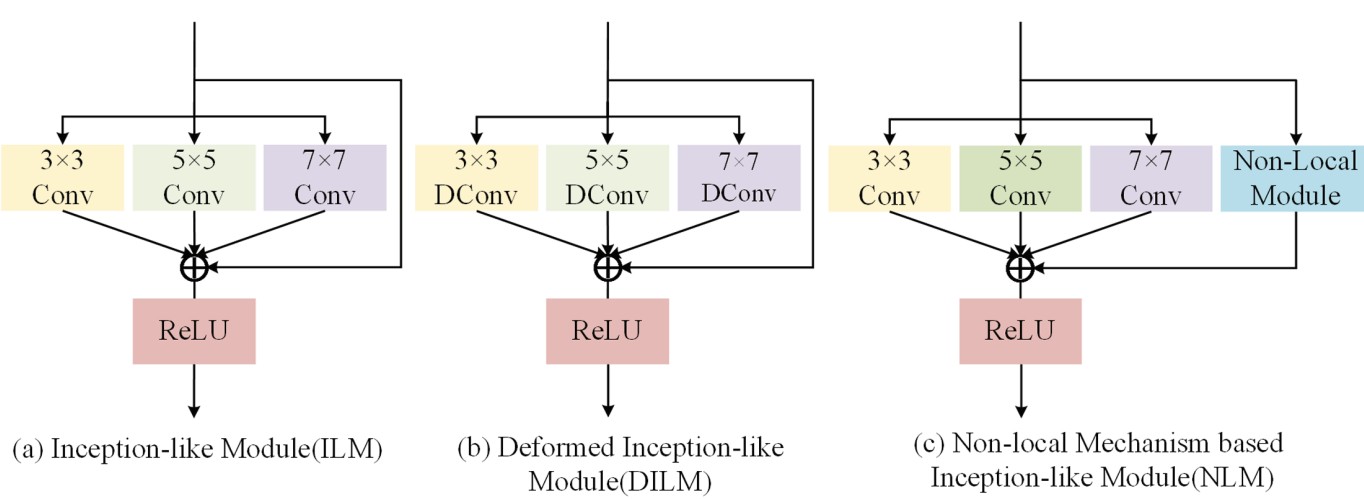

(a) Inception-like Module(ILM)

(b) Deformed Inception-like Module(DILM)

(c) Non-local Mechanism based Inception-like Module(NLM)

**Fig 2. Inception-like modules.** (**a**) Based on convolution with three different kernel sizes to expand the receptive field of PCN. (**b**) Employs deformable convolution to help GRN warp the sample grid flexibly. (**c**) Utilizes a non-local module in a residual branch to capture global information in the bottlenecks of both PCN and GRN.

where $w$ is the summation of the sampled values and $p_n$ enumerate the location in $\mathcal{R}$. The convolution kernel $\mathcal{R}$ determines the size of the receptive field. Traditional convolution operations slide over the image using a fixed kernel shape, with sampling points at predetermined positions within the kernel. This constraint limits the network's ability to adapt to object deformations within the image. To address this limitation and enhance the network's adaptability to targets with geometric distortions, we incorporate deformable convolution (DCN) [41]. DCN introduces learnable offsets that dynamically adjust the sampling locations of the convolution kernel, enabling it to better accommodate image deformations and complex structures. Specifically, in DCN, the convolution kernel $\mathcal{R}$ is augmented with offsets $\left\{\triangle p_n | n = 1, ..., N\right\}$ (N=$|\mathcal{R}|$), to enhance the modeling capability of the network for geometric transformations. This process can be formulated as:

$$y(p_0) = \sum_{p_n \in \mathcal{R}} w(p_n) \cdot x(p_0 + p_n + \triangle p_n) \tag{7}$$

To effectively incorporate the benefits of deformable convolution in addressing geometric transformations within the GRN, we have developed a multi-scale deformable convolution module, named deformable inception-like module (DILM), as illustrated in Fig. 2b. This module employs deformable convolution kernels with varying receptive fields to capture fine-grained local deformations and broader-scale distortion patterns simultaneously. In the DILM, grid information is deformed and mapped using 3×3, 5×5, and 7×7 convolutional kernels with learnable deformation biases. The extracted feature information is then combined with the input features in a residual manner. Let $B_{3\times3}$, $B_{5\times5}$ and $B_{7\times7}$ denote the deformation biases of the respective deformable convolution kernels, and let $\sigma$ represent the activation function. The operation of this Deformable Inception-like module can be formulated as follows:

$$DConv_{3\times3} : y_{3\times3} = DCN(x, B_{3\times3}) \tag{8}$$

$$DConv_{5\times5} : y_{5\times5} = DCN(x, B_{5\times5}) \tag{9}$$

$$DConv_{7\times7} : y_{7\times7} = DCN(x, B_{7\times7}) \tag{10}$$

Here we utilized $DCN$ to denote the deformable convolution, then the output of the multi-level fusion features in DILM are computed by Eq 7 as:

$$y = \sigma\left(y_{3\times3} + y_{5\times5} + y_{7\times7} + x\right) \tag{11}$$

This multi-scale structure effectively mitigates nonlinear geometric distortions across different scales, facilitating more accurate restoration of distorted projection images. By integrating the DILM with the non-local mechanism based module introduced later, our approach enhances the compensation of geometric deformations, yielding improved visual results. This effectiveness is validated through the visualization results of real surface compensation experiments.

**3.2.3 NLM: non-local mechanism based inception-like module** While the designed ILM and DILM enhance the perceptual awareness of the feature map, their ability to extract global information remains limited. To overcome this limitation, we integrate the non-local

attention mechanism [42], which enables the model to establish relationships between different positions on a global scale, thereby more effectively capturing long-range contextual information. By mapping features into distinct spaces and computing correlations between positions, attention weights are derived, encapsulating the importance of each position relative to others. These weights are then utilized to merge the original input features, culminating in a comprehensive global feature representation. Integrating this mechanism with the ILM reinforces the compensation network's ability to accumulate all-encompassing global insights from the feature map. Consequently, it establishes substantial constraints that promote global-scale textural coherence. This module is depicted in Fig 2c. Positioned at the network's bottleneck, this integration facilitates comprehensive global feature acquisition with a minimal increase in parameters. This augmentation significantly enhances the network's capacity for capturing long-range dependencies, thereby effectively extracting texture and geometric intricacies from input images.

**3.2.4 Loss function** For the design of the loss function, some research [3,43–46] have shown excellent performance with the combination of $\mathcal{L}_1$ loss and $\mathcal{L}_{\text{SSIM}}$. They are defined as:

$$\mathcal{L}_1 = \frac{1}{N} \sum_{i=1}^{N} |y_i - \hat{y}_i| \tag{12}$$

where $N$ represents the total number of pixels, $y_i$ represents the ground truth pixel value, and $\hat{y}_i$ represents the predicted pixel value. And

$$\mathcal{L}_{SSIM} = \frac{(2\mu_y\mu_{\hat{y}} + C_1)(2\sigma_{y\hat{y}} + C_2)}{(\mu_y^2 + \mu_{\hat{y}}^2 + C_1)(\sigma_y^2 + \sigma_{\hat{y}}^2 + C_2)} \tag{13}$$

where $y$ and $\hat{y}$ represent ground truth image and inferred image, respectively, $\mu_y$ and $\mu_{\hat{y}}$ represent the mean pixel values of the two images, $\sigma_y$ and $\sigma_{\hat{y}}$ represent the standard deviations of the two images, $\sigma_{y\hat{y}}$ represents the covariance of the two images, and $C_1$ and $C_2$ are two constants used to stabilize the denominator.

The $\mathcal{L}_1$ loss prioritizes pixel-level coherence, whereas the $\mathcal{L}_{SSIM}$ loss emphasizes structural and perceptual uniformity. Notably, subtle human-eye discernible enhancements in generated images could result in higher $\mathcal{L}_{SSIM}$ scores, despite potentially causing greater pixel-level disparities in the $\mathcal{L}_1$ loss, thus lowering PSNR. To simultaneously preserve perceptual quality and pixel-level intricacies, we introduce a local similarity loss ($\mathcal{L}_{ls}$). Through local patch comparisons, $\mathcal{L}_{ls}$ prevents error compensation between distinct regions, as observed with global $\mathcal{L}_1$ loss. By doing so, $\mathcal{L}_{ls}$ preserves the precision of $\mathcal{L}_1$ while minimizing the penalty imposed on perceptually enhanced regions. The collective employment of $\mathcal{L}_1$, $\mathcal{L}_{SSIM}$, and $\mathcal{L}_{ls}$ serves to strike a balance between perceptual quality and pixel-level consistency, leading to enhanced visual outcomes. Calculating the $\mathcal{L}_{ls}$ loss involves partitioning the image into local regions using fixed-sized windows. For each local region $y_i'$ and $\hat{y}_i'$, the $\mathcal{L}_1$ loss can be used to calculate the difference between them, denoted as $\mathcal{L}_1(y_i', \hat{y}_i')$. Then local similarity loss $\mathcal{L}_{ls}$ can be expressed as the average of the $\mathcal{L}_1$ losses of all local regions:

$$\mathcal{L}_{ls} = \frac{1}{N'} \sum_{i=1}^{N'} \mathcal{L}_1(y_i', \hat{y}_i') \tag{14}$$

To harness the optimal attributes of these loss functions, we aggregate them by summation:

$$\mathcal{L} = \mathcal{L}_1 + \mathcal{L}_{SSIM} + \mathcal{L}_{ls} \tag{15}$$

Incorporating the $\mathcal{L}_1$ loss, $\mathcal{L}_{SSIM}$, and $\mathcal{L}_{ls}$ loss within our network effectively mitigates the presence of blurred results, while concurrently preserving the overarching structure and intricate local details in the compensation task. This amalgamation culminates in improved image quality.

# 4 Experimets

This section provides the experimental evaluation of our proposed *IDNet* for geometric and photometric compensation. Experiments were conducted on two datasets: the benchmark dataset introduced by Huang et al. [3], and our proposed dataset captured using the pro-cam system under local environmental conditions. Section 4.1 introduces the details of these two datasets. Section 4.2 outlines the training details. Subsequently, Section 4.3 presents comprehensive experimental results. To analyze the impact of different modules and loss functions, ablation studies were shown in Section 4.4.

## 4.1 Datasets

**Compennet++ Full Compensation(CFC) dataset [2].** The Full Compensation dataset comprises 500 training images and 200 testing images, each possessing a resolution of 640x480 pixels. It comprises 20 unique combinations of pro-cam system positions and lighting conditions. Each of these combinations includes a dataset of images projected onto distinct surfaces.

**Large Full Compensation(LFC) Dataset.** To address the problems resulting from lack of data, e.g. network overfitting and suboptimal generalizability, we propose a Large Full Compensation dataset with abundant samples. This dataset, curated locally, consists of 5,000 images, with 4,000 images for training and 1,000 images for testing. We projected and captured these images on local projection surfaces at different positions and under different environmental lighting. All samples projected onto the local real surface were displayed using an Optoma EH412 projector and captured with a Nikon Z6II camera. During dataset acquisition, the projector fully projected images onto the projection surface, while the camera captured the entire projected image. The projector was positioned 1 meter from the projection surface, and the camera was placed 1.2 meters away. After capturing the projected images, the raw images were transformed to a resolution of 640×480 pixels.

## 4.2 Training details

The network was trained on two NVIDIA 3060 GPUs. We employed the AdamW optimizer, with the weight decay as $10^{-4}$. At the 800th iteration, the learning rate was scaled by a decay rate of 0.2. The overall training regimen spanned 1500 iterations in total.

## 4.3 Experimental results

In this section, we undertake a series of experiments to gain deeper insights into the impact of different networks on real projection compensation, as well as to assess the influence of the dataset on compensation outcomes. The compensation procedure is depicted in Fig 3. The trained network generates a compensated projector input image by utilizing both the surface

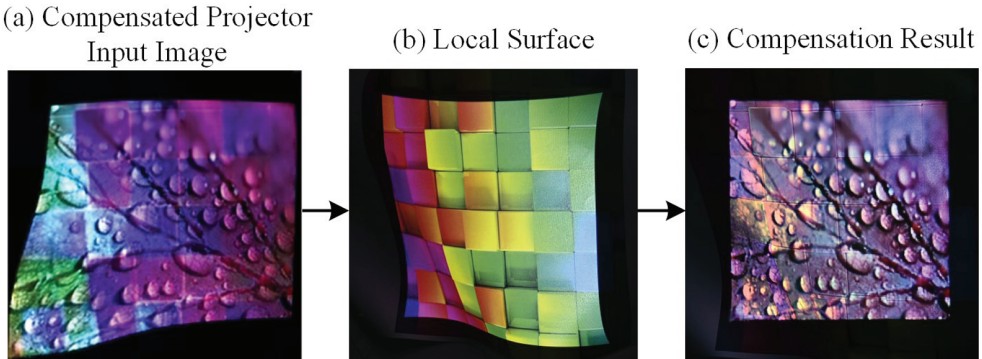

**Fig 3. Real surface compensation.** (**a**) Compensated image generated by the network for projector input. (**b**) Local real surface. (**c**) Results after projecting (a) onto (b).

**Table 2. Full compensation dataset experiment results.**

| Method | Surface1 | | | Surface2 | | | Surface3 | | |
|---|---|---|---|---|---|---|---|---|---|
| | PSNR ↑ | RMSE ↓ | SSIM ↑ | PSNR ↑ | RMSE ↓ | SSIM ↑ | PSNR ↑ | RMSE ↓ | SSIM ↑ |
| CompenNet++ | 24.8454 | 0.0991 | 0.8280 | 22.7473 | 0.1262 | 0.7462 | 22.4246 | 0.1310 | 0.7336 |
| CompenNeSt++ | 24.7951 | 0.0997 | 0.8352 | 23.1191 | 0.1209 | 0.7605 | 22.5391 | 0.1293 | 0.7347 |
| *IDNet* | **26.4865** | **0.0821** | **0.8611** | **23.9069** | **0.1105** | **0.7676** | **23.4752** | **0.1161** | **0.7413** |

image and the intended view image as inputs. Subsequently, this compensated projector input image is projected onto the local real surface, culminating in a full compensation result.

To assess the performance of the comparison methods and the designed modules, we used three common metrics: PSNR, RMSE, and SSIM. PSNR is a widely adopted metric for evaluating image quality, particularly in tasks like denoising, compression, and compensation. It measures the ratio between the maximum pixel value and the noise (or error) between the original and reconstructed images, with higher values indicating better quality. RMSE quantifies the average magnitude of reconstruction error by computing the square root of the mean squared differences between corresponding pixels. A lower RMSE indicates closer reconstruction to the original image, although it does not fully account for perceptual quality. For this reason, RMSE is often used alongside SSIM, which evaluates structural similarity based on luminance, contrast, and structural information. Unlike PSNR and RMSE, SSIM focuses on perceived image quality, with values ranging from 0 to 1, where 1 indicates perfect structural similarity. A higher SSIM suggests a more faithful reconstruction in terms of human visual perception, offering a more comprehensive measure of image quality.

**4.3.1 Local real surface compensation with CFC dataset** In this experiment, the training data is obtained based on the CFC dataset, which was projected onto three distinct local real surfaces characterized by their curvature and textural complexity. These surfaces exhibit varying degrees of color saturation, luminosity and geometric distortion. The first surface's dataset was captured under indoor environmental lighting, while the second and third surfaces were photographed under conditions of low ambient illumination. We compared three full compensation methods—CompenNet++, CompenNeSt++, and *IDNet*—on these captured images, evaluating their performance through comparative metrics and visual assessment. Table 2 summarizes the test results after training completion for each method, Fig 4 shows the training and testing curves of various methods on different surfaces, and Fig 5 illustrates the visual results of full compensation on real surfaces using different approaches.

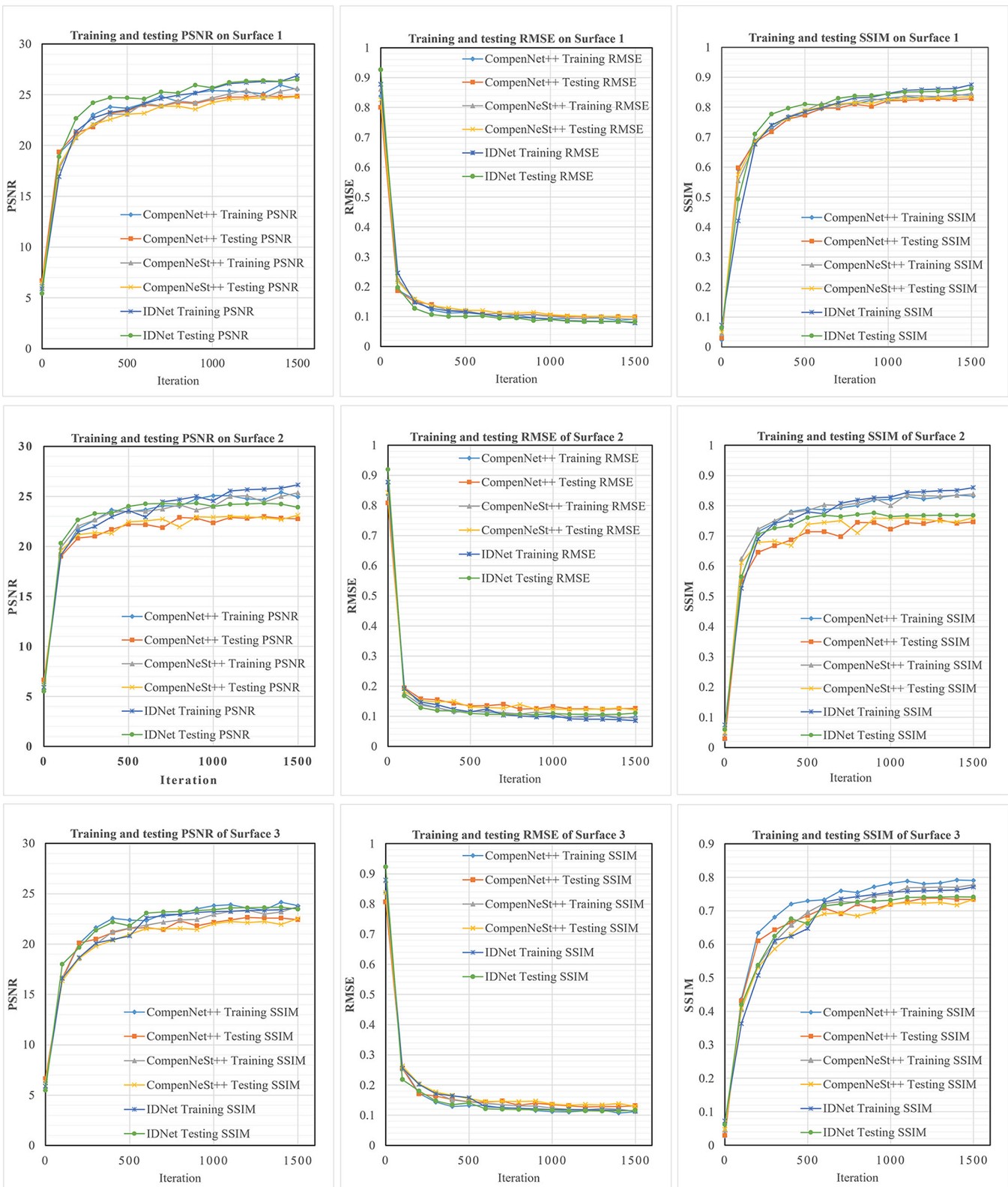

**Fig 4. Performance comparison of training and testing curves across different methods on CFC dataset.** Examining the training and testing curves on the CFC dataset, our proposed method consistently outperformed other competing approaches across virtually all stages of training and testing phases, demonstrating superior performance throughout the entire evaluation process.

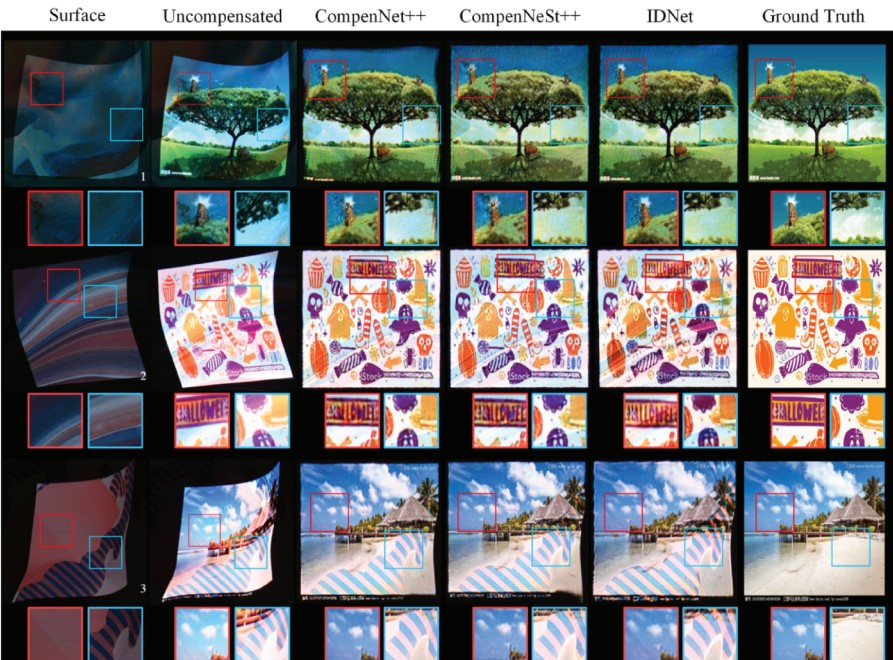

**Fig 5. Local real surface compensation with CFC dataset.** We project the CFC dataset onto the local real surfaces (**a**). The uncompensated effect are shown in (**b**), while (**c**)–(**e**) present the full compensation effect of different methods. (**f**) represents the ground truth images.

As demonstrated in Table 2, on the Surface 1, our approach outperforms all other methods. Specifically, our method demonstrates an 6.6% improvement in PSNR, a 20.7% reduction in RMSE compared to CompenNet++ and a 3.1% improvement in SSIM compared to CompenNeSt++. For the Surface 2, performance improvements vary across metrics, with our method achieving a 3.4% improvement in PSNR, a 9.4% reduction in RMSE, and a 0.9% improvement in SSIM compared to CompenNeSt++. Similarly, for the Surface 3, our method achieves a 4.2% improvement in PSNR, an 11.4% reduction in RMSE, and a 0.9% improvement in SSIM compared to CompenNeSt++.

Fig 4 presents a comprehensive comparison of training and testing metrics (PSNR, RMSE, and SSIM) curves between our *IDNet* and other methods. The results demonstrate several advantages of our approach: On Surface 1, *IDNet* achieves superior performance with both training and testing curves consistently above the baseline methods. The PSNR curves show that our method outperforms CompenNet++ and CompenNeSt++ in both training and testing phases. While all methods show rapid initial improvement, *IDNet* maintains higher PSNR values throughout the training process and achieves better final performance. The RMSE curves of our method are consistently lower than those of the other two methods, indicating that our method is more effective in reducing errors. The SSIM curves of our method are also higher than those of the other two methods, indicating that our method is more effective in preserving image structure. On Surface 2, our method achieves the best performance in terms of PSNR, RMSE, and SSIM. The curves of our method are more stable and converge faster than those of CompenNet++ and CompenNeSt++. This indicates that our method is more robust and efficient in learning the compensation task. On the most challenging Surface

3, *IDNet* demonstrates more robust performance. While all methods show decreased absolute performance due to surface complexity, *IDNet* maintains a more consistent relationship between training and testing curves. The RMSE curves particularly highlight this advantage, with *IDNet* achieving lower error rates in both training and testing scenarios compared to other methods.

In the real compensation results shown in the Fig 5, CompenNet++ and CompenNeSt++ exhibited significant edge distortion repair errors in their compensation results on surfaces 1 and 3. In the restoration of surface 2, all three comparative methods demonstrated good geometric correction effects. However, due to the presence of low-luminosity regions on this surface, it can be observed that all three methods suffered from insufficient color compensation. Similarly, on surface 3, with its striped regions, none of the methods achieved satisfactory compensation results. This performance is related to factors such as the light-absorbing properties of the surface material, low-luminosity colors, and the maximum brightness of the projector. Due to the incorporation of multiscale features in our designed ILMs, they can capture more local and global information. As a result, the compensation effect of *IDNet* outperforms the other two methods in restoring image texture structures and image edges, as presented in the global map of the images and detail bounding boxes.

**4.3.2 Local real surface compensation with LFC dataset** In this experiment, CompenNet++, CompenNeSt++, and our method *IDNet* are trained on the LFC dataset to generate compensated projector input images. We conducted full compensation comparison experiments on three Lambertian surfaces and one non-Lambertian surface. Among the Lambertian surfaces, Surface 1 and Surface 2 have relatively low color saturation, while Surface3 exhibits locally higher color saturation. Surface 4, as a non-Lambertian surface, features high reflectivity across its entire surface and includes localized black projection areas. As demonstrated in Table 3, Figs 6 and 7, our proposed method, *IDNet*, achieves superior performance on the LFC dataset in terms of both quantitative metrics and visual quality compared to CompenNet++ and CompenNeSt++.

As demonstrated in Table 3, *IDNet* shows consistent improvements across all evaluation metrics and surfaces. On Surface 1, IDNet outperforms previous best results with improvements of 3.3% in PSNR, 9.8% in RMSE, and 1.1% in SSIM, demonstrating advancements in image quality, error reduction, and structural fidelity. On Surface 2, *IDNet* continues to show advantages with improvements of 1.5% in PSNR, 4.3% in RMSE, and 0.9% in SSIM. The performance gains are particularly significant on Surface 3, where *IDNet* achieves substantial improvements of 8.2% in PSNR, 24.5% in RMSE, and 5.8% in SSIM. For Surface 4, *IDNet* maintains consistent performance improvements over CompenNeSt++, with gains of 2.0% in PSNR, 5.3% in RMSE, and 1.7% in SSIM.

Figure 6 presents the training and testing curves for different metrics across four surfaces of LFC dataset. The PSNR curves indicate that *IDNet* consistently outperforms both CompenNet++ and CompenNeSt++ across all surfaces. Notably, *IDNet* achieves higher PSNR values,

**Table 3. Multiple surfaces projection compensation.**

| Method | Surface1 | | | Surface2 | | | Surface3 | | | Surface4 | | |
|---|---|---|---|---|---|---|---|---|---|---|---|---|
| | PSNR ↑ | RMSE ↓ | SSIM ↑ | PSNR ↑ | RMSE ↓ | SSIM ↑ | PSNR ↑ | RMSE ↓ | SSIM ↑ | PSNR ↑ | RMSE ↓ | SSIM ↑ |
| CompenNet++ | 24.2269 | 0.1065 | 0.8264 | 23.7997 | 0.1118 | 0.8128 | 22.0515 | 0.1368 | 0.7635 | 22.4486 | 0.1307 | 0.7494 |
| CompenNeSt++ | 24.4980 | 0.1032 | 0.8307 | 23.6971 | 0.1132 | 0.8051 | 23.1603 | 0.1204 | 0.7862 | 22.2077 | 0.1343 | 0.7489 |
| textitIDNet | **25.3129** | **0.0940** | **0.8402** | **24.1681** | **0.1072** | **0.8198** | **25.0642** | **0.0967** | **0.8317** | **22.8984** | **0.1241** | **0.7619** |

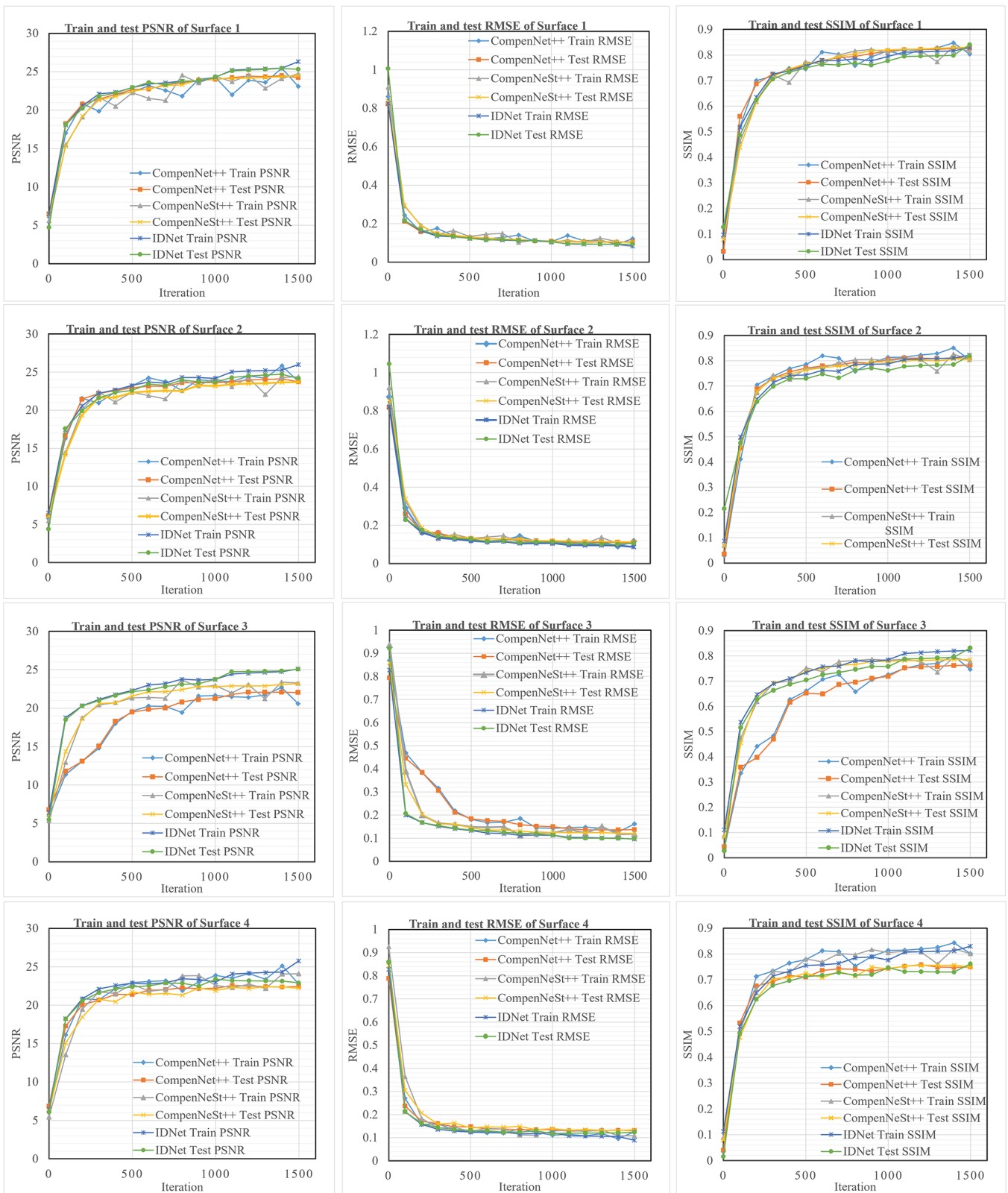

**Fig 6. Performance comparison of training and testing curves across different methods on LFC dataset.** From the training and testing curves of different methods, it can be observed that, overall, IDNet demonstrates more stable training and testing curves compared to other methods, with higher consistency in their trends. This indicates that IDNet possesses good stability and generalization capability, enabling it to effectively process unseen new data.

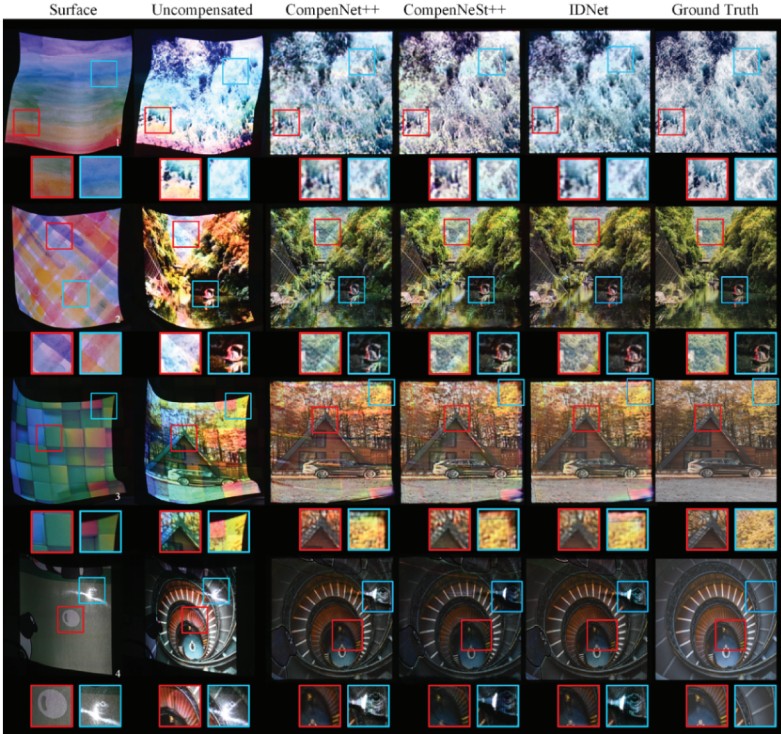

**Fig 7. Local real surface compensation on LFC dataset.** When the surface exhibits intense geometric distortion, other methods have difficulty properly aligning the compensated image during geometric correction, owing to their constrained receptive fields. In contrast, our proposed method IDNet is able to effectively address this challenge, attributable to its expanded receptive field.

particularly in the later stages of training, which indicates superior image quality reconstruction. The training and testing curves for *IDNet* show minimal gaps, suggesting excellent generalization and the ability to maintain high-quality reconstruction even on unseen data. In contrast, both CompenNet++ and CompenNeSt++ show significant discrepancies between training and testing PSNR values, indicating that these methods tend to overfit to the training data and fail to generalize effectively to the test set. The RMSE results further support the superiority of *IDNet*. Throughout the training process, *IDNet* exhibits the lowest RMSE values for both training and testing, which highlights its ability to minimize reconstruction errors. More importantly, the gap between training and testing RMSE curves for *IDNet* remains narrow, suggesting robust generalization capabilities. On the other hand, CompenNet++ and CompenNet display higher testing RMSE values, especially in the later iterations, indicating a lack of generalization and higher reconstruction errors on unseen data. Additionally, the larger gap between training and testing RMSE curves for these methods points to overfitting issues. The SSIM curves reinforce the findings from PSNR and RMSE evaluations. *IDNet* achieves the highest SSIM values, reflecting a better structural similarity between the original and reconstructed images. The consistency between the training and testing SSIM curves further emphasizes the robust performance of *IDNet* across different datasets. Conversely, CompenNet++ and CompenNet show lower SSIM values and greater discrepancies between their training and testing curves. This suggests that while these methods perform well on the training data, they fail to maintain the same level of quality when tested on unseen data.

Attending to the red and blue detail boxes as visualized in Fig 7, the results of *IDNet* in the last column demonstrate significantly better texture details and reduced color distortions compared to CompenNet++ and CompenNeSt++. More notably, *IDNet* excels in handling the severe distortions present in Surface 3, generating clean and sharp edges without noticeable warping artifacts. On the other hand, both CompenNet++ and CompenNeSt++ still exhibit substantial edge deformation errors, particularly in the more complex regions of the image. This enhanced performance can be attributed to the Inception-like modules in *IDNet*, which extend the receptive field to capture both local features and global context. By incorporating information from larger image regions, *IDNet* is able to reconstruct the projected image with greater accuracy, particularly in minimizing boundary distortions. On Surface 4, where the projection surface exhibits strong reflective properties, most methods succeed in achieving effective photometric and geometric compensation in non-specular reflection areas. However, in regions with specular reflections, as highlighted in the blue detail box, none of the methods can provide satisfactory compensation results. Furthermore, when black textured regions are present on the projection surface, all compensation methods face challenges in fully compensation for the distortions, as the light-absorbing nature of black affects the compensation process. Among these methods, CompenNeSt++ exhibits the most significant loss of detail in black compensation areas, and it fails to effectively address the geometric distortion along the edges. In contrast, both CompenNet++ and *IDNet* demonstrate superior detail recovery in these regions, with *IDNet* particularly excelling in maintaining image clarity and structure.

In summary, the proposed *IDNet* excels in both geometric and photometric compensation through its innovative integration of multi-scale deformable convolution and non-local attention mechanism. The compensated images demonstrate superior visual quality with enhanced clarity and refined edge integrity, while achieving state-of-the-art performance in geometric and photometric recovery. Both qualitative and quantitative evaluations confirm *IDNet*'s exceptional capability in delivering full compensation, effectively addressing both geometric distortions and photometric inconsistencies.

## 4.4 Ablation study

The ablation study experiments were conducted to analyze the impact of designed modules and loss functions. Section 4.4.1 presents the ablation study for each proposed module on the CFC dataset. Section 4.4.2 shows the ablation study of loss functions on surface 3 of the LFC dataset.

**4.4.1 Module ablation**  To validate the effectiveness of our proposed modules, we conducted an ablation study comparing the baseline network to various versions with additional components. The baseline architecture comprised ILM, convolutional and transposed convolutional layers, and residual modules only. The DILM version replaced the ILMs in the GRN with our proposed DILMs. Finally, the full model added the NLM modules to the bottleneck of both the geometric and photometric compensation networks.

Experiments were performed on the CFC dataset, as the results in Table 4 reflecting averages over 20 subsets. From the visualization in Fig 8, adding the DILM and NLM modules consistently improves compensation performance. The DILM version reduces distortion errors by expanding the receptive field to incorporate more global context. The NLM model further enhances the structural feature extraction capability, capturing more fine-grained details.

**Table 4. Ablation study of the designed modules.**

| Baseline | DILM | NLM | PSNR ↑ | RMSE ↓ | SSIM ↑ |
|---|---|---|---|---|---|
| ✓ | | | 22.0254 | 0.1386 | 0.7459 |
| ✓ | ✓ | | 22.3243 | 0.1345 | 0.7550 |
| ✓ | | ✓ | 22.1321 | 0.1370 | 0.7630 |
| ✓ | ✓ | ✓ | **22.4449** | **0.1322** | **0.7732** |

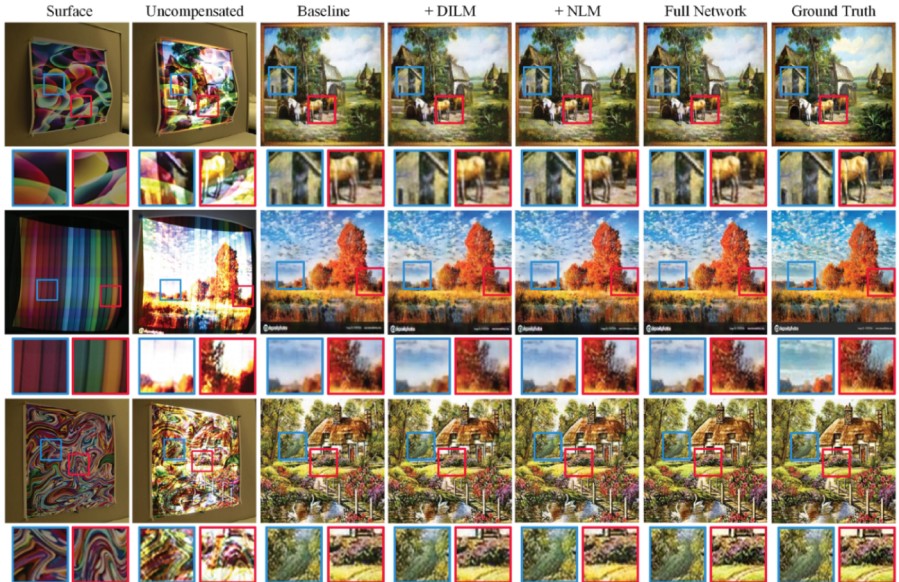

**Fig 8. Ablation of modules.** With the incorporation of either DILM or NLM, the network attained enhanced compensation for image details. When compensated using the full network, the resulting image became clearer and texture details were enriched.

For the quantitative results shown in Table 4, adding the DILM module to the baseline leads to gains of 1.4% in PSNR, 0.5% in RMSE and 1% in SSIM, respectively. Incorporating the NLM further improved performance by 0.5% for PSNR, 1.2% in RMSE and 2.3% for SSIM. With both modules, *IDNet* achieved the state-of-the-art performance.

While each module contributes individually, strategically integrating DILM and NLM at different network stages has a synergistic effect. The DILM initially reduces distortion by expanding the receptive field, while the NLM subsequently enhances feature extraction to capture finer structural textures. These modules working cooperatively to improve the compensation accuracy. The ablation study further confirms that the complete *IDNet* architecture, incorporating both DILM and NLM, achieves superior performance in compensating for projection.

**4.4.2 Ablating the losses components** Through a systematic ablation study, we analyzed the impact of various loss formulations on the image restoration performance of our proposed model as quantified by metrics including PSNR, RMSE, and SSIM in Table 5. The results clearly demonstrate the complementary benefits of combining multiple loss terms. Using $\mathcal{L}_1$ loss alone achieves reasonable performance with a PSNR of 22.87 dB, RMSE of 0.1245, and SSIM of 0.7397. In comparison, $\mathcal{L}_2$ loss alone shows slightly inferior performance across all

Table 5. Ablation study of different loss functions.

| Loss Function | PSNR ↑ | RMSE ↓ | SSIM ↑ |
|---|---|---|---|
| $\mathcal{L}_1$ | 22.8675 | 0.1245 | 0.7397 |
| $\mathcal{L}_2$ | 22.4434 | 0.1307 | 0.6847 |
| $\mathcal{L}_1 + \mathcal{L}_{SSIM}$ | 23.3097 | 0.1183 | 0.7925 |
| $\mathcal{L}_1 + \mathcal{L}_{ls}$ | 22.6529 | 0.1276 | 0.7238 |
| $\mathcal{L}_1 + \mathcal{L}_{SSIM} + \mathcal{L}_{ls}$ | **25.0642** | **0.0967** | **0.8317** |

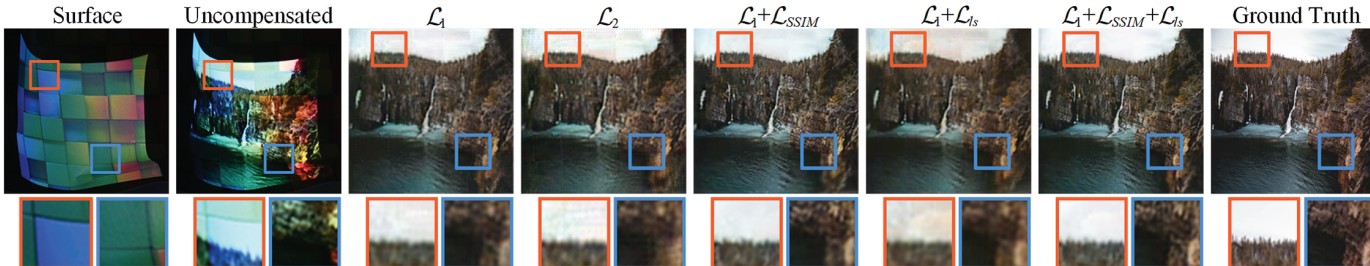

**Fig 9. Visualization of the ablation study over loss functions.**

metrics, suggesting that $\mathcal{L}_1$ loss is more suitable for our full compensation task. When combining $\mathcal{L}_1$ loss with $\mathcal{L}_{SSIM}$ loss, we observe notable improvements across all metrics, achieving a PSNR of 23.31 dB, RMSE of 0.1183, and SSIM of 0.7925. This improvement indicates that the $\mathcal{L}_{SSIM}$ loss effectively enhances the structural fidelity of the compensated images. The combination of $\mathcal{L}_1$ and $\mathcal{L}_{ls}$ loss shows moderate improvement over using $\mathcal{L}_1$ loss alone, but the gains are less significant compared to the $\mathcal{L}_1 + \mathcal{L}_{SSIM}$ combination. Most notably, the full combination of all three losses ($\mathcal{L}_1 + \mathcal{L}_{SSIM} + \mathcal{L}_{ls}$) achieves the best performance by a substantial margin, with a PSNR of 25.06 dB, RMSE of 0.0967, and SSIM of 0.8317. These results represent significant improvements of approximately 9.6% in PSNR, 28.7% in RMSE, and 12.4% in SSIM compared to using $\mathcal{L}_1$ loss alone. This demonstrates that each loss term contributes unique and complementary supervision signals, leading to more effective photometric and geometric compensation. By synergistically integrating these complementary components, our model achieved enhanced performance in compensating for fine-grained texture while preserving strong PSNR and structural similarities.

As shown in the qualitative visualization in Fig 9, using only the $\mathcal{L}_1$ or $\mathcal{L}_2$ loss results in uneven texture restoration, as well as residual traces of color and texture interference from the projected surface. These issues are effectively addressed by applying the $\mathcal{L}_1 + \mathcal{L}_{SSIM}$ and the comprehensive $\mathcal{L}_1 + \mathcal{L}_{SSIM} + \mathcal{L}_{ls}$ loss. Importantly, the later approach produces significantly clearer details in the compensated images, surpassing the outcomes achieved by other methods.

## Conclusions

In this paper, we propose a new full-compensation network that excels at compensating geometric and photometric distortions, denoted as *IDNet*. This achievement is facilitated through the incorporation of three tailored Inception-like modules, namely ILM, DILM and NLM. The ILM extracts multiscale features by employing convolutional kernels of varying sizes. The DILM achieves more flexible geometric transformations of the warping grid through deformation convolutional operations. Moreover, the NLM aids the entire network in acquiring

more global information, thereby enabling it to perceive a broader view of features. In the experiments, we demonstrate that *IDNet* is superior to the existing methods when it came to fully compensating geometric distortions on projected surfaces. The adept compensation of edges from *IDNet*, where existing the significant distortions, highlights the importance of leveraging global information and multiscale receptive fields. By capitalizing on a holistic view of the projected imagery, our model can effectively reconstruct degraded edges and contours.

## Limitations

Our experimental findings reveal a critical limitation of the proposed method: its reduced effectiveness in achieving geometric compensation on surfaces exhibiting severe geometric distortions. To mitigate this challenge, our future research will explore the implementation of multi-projector systems, which can potentially provide more comprehensive compensation strategies for structurally complex surfaces. Moreover, within photometric modeling, surface normal information serves as a fundamental parameter in characterizing a surface's intricate optical properties, including reflection, refraction, and light scattering. Inspired by notable research on normal estimation techniques [7–9], we aim to explore the potential integration of precise normal information into our photometric and geometric correction frameworks. By integrating insights from normal estimation methodologies with our current approach, we hope to enhance the robustness and visual quality of projection compensation results.

## Author contributions

**Conceptualization:** Yuqiang Zhang, Huamin Yang, Cheng Han, Chao Xu, Shiyu Lu.

**Data curation:** Yuqiang Zhang.

**Formal analysis:** Yuqiang Zhang.

**Funding acquisition:** Huamin Yang, Cheng Han, Chao Zhang.

**Investigation:** Yuqiang Zhang.

**Methodology:** Yuqiang Zhang.

**Project administration:** Yuqiang Zhang.

**Resources:** Yuqiang Zhang.

**Software:** Yuqiang Zhang.

**Supervision:** Yuqiang Zhang.

**Validation:** Yuqiang Zhang.

**Visualization:** Yuqiang Zhang.

**Writing – original draft:** Yuqiang Zhang.

**Writing – review & editing:** Yuqiang Zhang.

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
