## [Decision Letter · Decision Letter 0]

22 Oct 2024

PONE-D-24-43381IDNet: An Inception-like Deformable Non-local Network for Projection Compensation over Non-flat Textured SurfacesPLOS ONE

Dear Dr. Yang,

Thank you for submitting your manuscript to PLOS ONE. After careful consideration, we feel that it has merit but does not fully meet PLOS ONE’s publication criteria as it currently stands. Therefore, we invite you to submit a revised version of the manuscript that addresses the points raised during the review process.

We look forward to receiving your revised manuscript.

Kind regards,

Yakun Ju

Academic Editor

PLOS ONE

“This research was funded by Jilin Province Science and Technology Development Plan Project, grant number: (20230101179JC) National Natural Science Foundation of China Youth Fund Project, grant number: (No. 61702051), Jilin Province Science and Technology Development Plan Project, grant number: (20200403188SF).”

“This research was funded by Jilin Province Science and Technology Development Plan Project, grant number: (20230101179JC) National Natural Science Foundation of China Youth Fund Project, grant number: (No. 61702051), Jilin Province Science and Technology Development Plan Project, grant number: (20200403188SF).”

“This research was funded by Jilin Province Science and Technology Development Plan Project, grant number: (20230101179JC) National Natural Science Foundation of China Youth Fund Project, grant number: (No. 61702051), Jilin Province Science and Technology Development Plan Project, grant number: (20200403188SF).”

5. We note that your Data Availability Statement is currently as follows: [All relevant data are within the manuscript and its Supporting Information files.]

6. PLOS requires an ORCID iD for the corresponding author in Editorial Manager on papers submitted after December 6th, 2016. Please ensure that you have an ORCID iD and that it is validated in Editorial Manager. To do this, go to ‘Update my Information’ (in the upper left-hand corner of the main menu), and click on the Fetch/Validate link next to the ORCID field. This will take you to the ORCID site and allow you to create a new iD or authenticate a pre-existing iD in Editorial Manager.

Additional Editor Comments:

Please check the reviewer's suggestions and provide one-by-one feedback.

Reviewers' comments:

Reviewer's Responses to Questions

**Comments to the Author**

1. Is the manuscript technically sound, and do the data support the conclusions?

Reviewer #1: Yes

Reviewer #2: Yes

2. Has the statistical analysis been performed appropriately and rigorously? 

Reviewer #1: Yes

Reviewer #2: N/A

3. Have the authors made all data underlying the findings in their manuscript fully available?

Reviewer #1: Yes

Reviewer #2: Yes

4. Is the manuscript presented in an intelligible fashion and written in standard English?

Reviewer #1: Yes

Reviewer #2: Yes

5. Review Comments to the Author

Reviewer #1: 1) The abstract is not expressing the novelty of the proposed approach. The whole abstract is not impressive and needs to be rewritten. Should focus on what is problem, why it is important to be solved. How it is solved and what are findings.

Reviewer #2: The manuscript proposes a novel deep learning method aimed at projection compensation over non-flat textured surfaces, achieving progress in both geometric and photometric compensation. However, there are several aspects that require improvement, and I recommend a major revision. Please check the attached for the detailed suggestions.

6. PLOS authors have the option to publish the peer review history of their article (what does this mean?). If published, this will include your full peer review and any attached files.

Reviewer #1: No

Reviewer #2: No

---

## [Author Response · Author response to Decision Letter 1]

5 Dec 2024

Dear Reviewer1,

Thank you for your valuable feedback on the abstract. We have revised it to better highlight the problem we are addressing and the novelty of our approach. The updated abstract focuses on the challenges of projector compensation on non-flat textured surfaces, particularly in image edge regions, where conventional convolution-based methods struggle. We propose IDNet, which integrates multi-scale deformable convolution modules and non-local attention mechanisms to effectively handle geometric distortions and capture global contextual information. Our experimental results demonstrate that IDNet achieves superior visual quality, particularly in challenging edge regions, while maintaining compensatory performance comparable to existing methods.

We believe this revision more clearly communicates the core contributions of our research. Thank you again for your helpful suggestion.

Best regards

Dear Reviewer2,

Thank you for your valuable feedback. In response to your comments, we have uploaded a corresponding response letter in the system. Based on your suggestions, we have made revisions to the manuscript. We sincerely appreciate your help in improving the quality of the manuscript.

Best regards

---

## [Decision Letter · Decision Letter 1]

20 Dec 2024

PONE-D-24-43381R1IDNet: An Inception-like Deformable Non-local Network for Projection Compensation over Non-flat Textured SurfacesPLOS ONE

Dear Dr. Yang,

Thank you for submitting your manuscript to PLOS ONE. After careful consideration, we feel that it has merit but does not fully meet PLOS ONE’s publication criteria as it currently stands. Therefore, we invite you to submit a revised version of the manuscript that addresses the points raised during the review process.

We look forward to receiving your revised manuscript.

Kind regards,

Yakun Ju

Academic Editor

PLOS ONE

Journal Requirements:

Reviewers' comments:

Reviewer's Responses to Questions

**Comments to the Author**

1. If the authors have adequately addressed your comments raised in a previous round of review and you feel that this manuscript is now acceptable for publication, you may indicate that here to bypass the “Comments to the Author” section, enter your conflict of interest statement in the “Confidential to Editor” section, and submit your "Accept" recommendation.

Reviewer #1: All comments have been addressed

2. Is the manuscript technically sound, and do the data support the conclusions?

Reviewer #1: Yes

3. Has the statistical analysis been performed appropriately and rigorously? 

Reviewer #1: Yes

4. Have the authors made all data underlying the findings in their manuscript fully available?

Reviewer #1: Yes

5. Is the manuscript presented in an intelligible fashion and written in standard English?

Reviewer #1: Yes

6. Review Comments to the Author

Reviewer #1: Introduction section is too short and written just like an abstract. It does not satisfy the need and requirements of an interested reader. Therefore author is suggested to rewrite the introduction section which should depicts the introduction of the field, existing methodologies and their associated issues, how the research gap is identified, your proposed approach adopts which unique technique which improves the it performance and make it novel and the contribution of their proposed approach. Author should use bullet 1 for the introduction section and so on for subsequent sections.

The bullet of the section Related work should be 2 and there some be some line of text between the main and subsequent heading. This suggestion should be followed in whole paper. Use the reference of the latest approaches of 2023-24 and then address the novelty of the proposed approach.

Figure 1,2,3,4 and 5 are missing within the manuscript.

Use a consistent formatting style for tables within the paper.

Results section needs major revision which demonstrates the training as well the testing results. In the meantime the impacts of each consider parameter in a detailed manner which improves the validity of the approach and quality of the manuscript.

7. PLOS authors have the option to publish the peer review history of their article (what does this mean?). If published, this will include your full peer review and any attached files.

Reviewer #1: No

---

## [Author Response · Author response to Decision Letter 2]

18 Jan 2025

Respond to Reviewer 1’s Comments

Suggestion 1:

Introduction section is too short and written just like an abstract. It does not satisfy the need and requirements of an interested reader. Therefore author is suggested to rewrite the introduction section which should depicts the introduction of the field, existing methodologies and their associated issues, how the research gap is identified, your proposed approach adopts which unique technique which improves the it performance and make it novel and the contribution of their proposed approach. Author should use bullet 1 for the introduction section and so on for subsequent sections.

Response:

Thank you for your valuable feedback.

We have revised the Introduction section in accordance with your suggestions. The updated version now provides a more comprehensive overview of the field, outlines existing methodologies along with their associated challenges, and highlights the identified research gap. Additionally, we have clearly explained the unique techniques adopted in our proposed approach, how they contribute to performance improvement, and what makes our approach novel. We believe these revisions better address the interests of readers and fulfill the requirements for a robust introduction. We have also organized the content using bullet points for clarity, as per your recommendation.

Suggestion 2:

The bullet of the section Related work should be 2 and there some be some line of text between the main and subsequent heading. This suggestion should be followed in whole paper. Use the reference of the latest approaches of 2023-24 and then address the novelty of the proposed approach.

Response:

Thank you for your helpful suggestion.

We have reviewed the latest paper from 2023-2024 and incorporated relevant approaches, along with their associated techniques and limitations, into the 'Related work' section. The section now reflects the most recent advancements in the field, and we have addressed how our proposed approach overcomes the challenges highlighted in these works, thereby emphasizing its novelty. Additionally, we have followed your formatting recommendation by adding appropriate spacing between the main and subsequent headings, ensuring consistency throughout the paper.

Suggestion 3:

Figure 1,2,3,4 and 5 are missing within the manuscript.

Response:

Thank you for pointing out the issue regarding the missing figures. According to the submission guidelines provided by PLOS One (https://journals.plos.org/plosone/s/submission-guidelines#loc-figures-and-tables), figures should be uploaded as separate files rather than being included within the manuscript itself. In line with these guidelines, Figures 1, 2, 3, 4, and 5 have been uploaded as individual files to the submission system. We hope this resolves the issue, and we appreciate your understanding.

Suggestion 4:

Use a consistent formatting style for tables within the paper.

Response:

Thank you for your valuable suggestion.

We have carefully reviewed the formatting of all tables in the manuscript and have made the necessary adjustments to ensure a consistent style throughout. The tables now follow a uniform formatting style, in line with the journal’s guidelines. We appreciate your attention to detail and believe these changes improve the clarity and presentation of the manuscript.

Suggestion 5:

Results section needs major revision which demonstrates the training as well the testing results. In the meantime the impacts of each consider parameter in a detailed manner which improves the validity of the approach and quality of the manuscript.

Response:

Dear Reviewer,

Thank you for your valuable comments regarding our manuscript. We fully agree with your suggestion that the Results section needs major revision to demonstrate both training and testing results.

Regarding the training and testing data, we need to candidly explain that due to technical issues (system reinstallation), some of our original training data and curves were not properly preserved. To ensure the reliability and completeness of our research, we have taken the following remedial actions:

1. Re-conducted part of the experimental data collection

2. Fully documented the new training and testing processes

3. Supplemented detailed comparative experimental data in sections 4.3.1 and 4.3.2 of the manuscript

4. Added comprehensive analysis of both training and testing curves

Through these revisions, we have:

• More systematically demonstrated the performance of our proposed method

• Strengthened the reliability of our experimental results

• Provided in-depth analysis of how each parameter impacts system performance

These modifications have not only enhanced the academic quality of our manuscript but have also provided us with valuable research experience. We sincerely appreciate your constructive feedback, which has played a crucial role in improving the quality of our paper. All changes in the revised manuscript have been highlighted in yellow for your convenience.

We are more than willing to make further improvements if you have any additional suggestions or concerns.

Best regards

---

## [Decision Letter · Decision Letter 2]

22 Jan 2025

IDNet: An Inception-like Deformable Non-local Network for Projection Compensation over Non-flat Textured Surfaces

PONE-D-24-43381R2

Dear Dr. Yang,

We’re pleased to inform you that your manuscript has been judged scientifically suitable for publication and will be formally accepted for publication once it meets all outstanding technical requirements.

Kind regards,

Yakun Ju

Academic Editor

PLOS ONE

Additional Editor Comments (optional):

Reviewers' comments:

Reviewer's Responses to Questions

**Comments to the Author**

1. If the authors have adequately addressed your comments raised in a previous round of review and you feel that this manuscript is now acceptable for publication, you may indicate that here to bypass the “Comments to the Author” section, enter your conflict of interest statement in the “Confidential to Editor” section, and submit your "Accept" recommendation.

Reviewer #1: All comments have been addressed

2. Is the manuscript technically sound, and do the data support the conclusions?

Reviewer #1: Yes

3. Has the statistical analysis been performed appropriately and rigorously? 

Reviewer #1: Yes

4. Have the authors made all data underlying the findings in their manuscript fully available?

Reviewer #1: (No Response)

5. Is the manuscript presented in an intelligible fashion and written in standard English?

Reviewer #1: Yes

6. Review Comments to the Author

Reviewer #1: (No Response)

7. PLOS authors have the option to publish the peer review history of their article (what does this mean?). If published, this will include your full peer review and any attached files.

Reviewer #1: No

---

## [Editor Report · Acceptance letter]

PONE-D-24-43381R2

PLOS ONE

Dear Dr. Yang,

I'm pleased to inform you that your manuscript has been deemed suitable for publication in PLOS ONE. Congratulations! Your manuscript is now being handed over to our production team.

Kind regards,

on behalf of

Dr. Yakun Ju

Academic Editor

PLOS ONE